# MiRNA Differences Related to Treatment-Resistant Schizophrenia

**DOI:** 10.3390/ijms24031891

**Published:** 2023-01-18

**Authors:** Daniel Pérez-Rodríguez, Maria Aránzazu Penedo, Tania Rivera-Baltanás, Tonatiuh Peña-Centeno, Susanne Burkhardt, Andre Fischer, José M. Prieto-González, José Manuel Olivares, Hugo López-Fernández, Roberto Carlos Agís-Balboa

**Affiliations:** 1NeuroEpigenetics Lab, Instituto de Investigación Sanitaria de Santiago (IDIS), Complejo Hospitalario Universitario de Santiago, 15706 Santiago de Compostela, Spain; 2Translational Neuroscience Group, Galicia Sur Health Research Institute (IIS Galicia Sur), Área Sanitaria de Vigo-Hospital Álvaro Cunqueiro, SERGAS-UVIGO, CIBERSAM-ISCIII, 36213 Vigo, Spain; 3Grupo de Neurofarmacología de Las Adicciones y Los Trastornos Degenerativos (NEUROFAN), Universidad CEU San Pablo, 28925 Madrid, Spain; 4Department for Epigenetics and Systems Medicine in Neurodegenerative Diseases, German Center for Neurodegenerative Diseases, 37075 Göttingen, Germany; 5Servicio de Neurología, Hospital Clínico Universitario de Santiago, 15706 Santiago de Compostela, Spain; 6Grupo Trastornos del Movimiento, Instituto de Investigación Sanitaria de Santiago (IDIS), Complejo Hospitalario Universitario de Santiago, 15706 Santiago de Compostela, Spain; 7Department of Psychiatry, Área Sanitaria de Vigo, 36312 Vigo, Spain; 8SING Research Group, Galicia Sur Health Research Institute (IIS Galicia Sur), SERGAS-UVIGO, 36213 Vigo, Spain; 9CINBIO, Department of Computer Science, ESEI-Escuela Superior de Ingeniería Informática, Universidade de Vigo, 32004 Ourense, Spain

**Keywords:** treatment resistant schizophrenia, microRNA, biomarkers, p53

## Abstract

Schizophrenia (SZ) is a serious mental disorder that is typically treated with antipsychotic medication. Treatment-resistant schizophrenia (TRS) is the condition where symptoms remain after pharmacological intervention, resulting in long-lasting functional and social impairments. As the identification and treatment of a TRS patient requires previous failed treatments, early mechanisms of detection are needed in order to quicken the access to effective therapy, as well as improve treatment adherence. In this study, we aim to find a microRNA (miRNA) signature for TRS, as well as to shed some light on the molecular pathways potentially involved in this severe condition. To do this, we compared the blood miRNAs of schizophrenia patients that respond to medication and TRS patients, thus obtaining a 16-miRNA TRS profile. Then, we assessed the ability of this signature to separate responders and TRS patients using hierarchical clustering, observing that most of them are grouped correctly (~70% accuracy). We also conducted a network, pathway analysis, and bibliography search to spot molecular pathways potentially altered in TRS. We found that the response to stress seems to be a key factor in TRS and that proteins p53, SIRT1, MDM2, and TRIM28 could be the potential mediators of such responses. Finally, we suggest a molecular pathway potentially regulated by the miRNAs of the TRS profile.

## 1. Introduction

Schizophrenia (SZ) is a heterogeneous psychiatric disorder that affects around 1% of the world’s population [1], and it is characterised by chronic psychotic symptoms and psychosocial impairment. Since the development of chlorpromazine in the 1950s, treatment for SZ is based on antipsychotic medications; however, about 20 to 50 percent of patients do not experience an improvement in their symptoms [2,3,4,5], which is known as treatment-resistant schizophrenia (TRS). The clinical definition of TRS is based on the failure of response to at least two sequential non-clozapine antipsychotic trials of sufficient dose, duration, and adherence [6]. Patients with TRS have higher rates of substance abuse, early cognitive decline, and higher rates of suicide ideation [7,8,9,10]. As pharmacological and therapeutic approaches differ between SZ and TRS [11,12,13,14], early identification of the resistance condition is essential for a rapid and effective pharmacological intervention, limiting damage to the patient and their environment and improving the adherence to treatment [15]. As a consequence, in the last two decades, an increasing number of studies have made great efforts to find molecular mechanisms behind TRS and spot biomarkers for clinical use. However, the difficult access to neural tissues, the subtle definition of TRS, and the lack of standardization in the bioinformatic methodologies hindered such endeavors [15,16,17,18,19].

In the last decade, micro-RNAs (miRNAs) emerged as promising biomarkers for mental conditions [16,20]. These molecules are small non-coding RNAs involved in the post-transcriptional regulation of gene expression. Their presence in almost all biological processes makes them easy to sample, whereas their susceptibility to everyday events, such as sleep, stress, or medications, gives them potential as biomarkers [21]. In this study, we employed NGS sequencing to characterise the complete blood miRNAome of schizophrenia patients differing in their response to antipsychotic treatment. To carry out the bioinformatics analyses, first we employed the highly replicable pipeline “myBrain-Seq v0.1.0” [22,23] for raw data preprocessing and differential expression analysis for defining a miRNA profile associated with the TRS condition (TRS profile). Then, we performed a hierarchical clustering analysis for exploring the sample grouping based on the TRS profile, using the R package hclust [24], and a functional analysis to explore some potential biological implications, using Cytoscape v3.9.1 [25] and StringApp v2.0.0 [26] tools. Our aim is twofold: first, to find a miRNA signature involved in TRS and second, to suggest its molecular roles and biological context. As a result, we found 16 miRNAs potentially involved in poor response to antipsychotic medication. Using this 16-miRNA signature, we were able to accurately classify more than two-thirds of our SZ and TRS samples and propose a molecular pathway in which these miRNAs could be involved, thus offering an interesting direction in which future research on TRS could be oriented.

## 2. Results

### 2.1. Profile of 16 miRNAs for Antipsychotic Resistance

The quality of the samples, the sequencing depth, and the proportion of assigned and mapped sequences can be consulted in Appendix A. Differential expression analysis of the miRNAome of schizophrenia patients with a normal antipsychotic response (MR) vs. schizophrenia patients with antipsychotic resistance (MNR) shows a total of 16 differentially expressed miRNAs potentially related to antipsychotic resistance (TRS profile), including 6 miRNAs whose expression was upregulated and 10 miRNAs that were downregulated. As shown in Figure 1A, the miRNAs differentially expressed in the resistant schizophrenia, compared to non-resistance schizophrenia, are miR-3127-3p (↑), miR-504-5p (↑), miR-3605-3p (↑), miR-6747-3p (↑), miR-3615 (↑), miR-1343-3p (↑), miR-145-5p (↓), miR-500a-3p (↓), miR-210-3p (↓), miR-199a-5p (↓), miR-548ay-3p (↓), miR-548ac (↓), miR-296-5p (↓), miR-660-5p (↓), miR-30b-5p (↓), and miR-223-3p (↓).

### 2.2. Hierarchical Clustering Analysis Using the TRS Profile

As we described in materials and methods, hierarchical clustering was performed twice: using samples in both MR and MNR groups and using samples in the MR, MNR, and first psychotic episode at hospital arrival (PA) groups. We assessed the Positive and Negative Syndrome Scale (PANSS) and the Self-assessment Anhedonia Scale (SAAS) scores as a way to improve the discriminative power of the TRS profile (Figure 1B); however, they show a high overlap between groups and, therefore, were discarded.

In the first dendrogram of MR and MNR samples (Figure 1C), two groups were differentiated. Cluster 1 had 73.2% of the MR samples, and cluster 2 had 72.4% of the MNR samples.

In the second dendrogram of the MR, MNR, and PA samples (Figure 1D), 3 groups were differentiated: in cluster 1, 73.81% of samples were MR; in cluster 2, 59.38% of samples were MNR; and in cluster 3, 55.6% were MNR samples. Regarding the PA samples, four of them fell in cluster 1, nine in cluster 2, and none of them fell in cluster 3. Cluster 2 had 7 of the 9 PA_R_ samples (77.78%) and 1 PA_NR_ (22.22%). Cluster 1 had the remaining 2 PA_R_ and PA_NR_ samples (50% each). If we remove PA samples, cluster 1 has 78.9% of the MR samples, and cluster 2 has 73.9% of the MNR samples. From now on, we will refer to cluster 1 as “MR cluster” and to cluster 2 as “MNR cluster”.

### 2.3. Functional Analysis

The first step of the functional analysis was the target prediction. Tarbase v8 [27] predicted 13,267 targets for the 16 miRNAs of the miRNA signature. Four miRNAs, namely, hsa-miR-548ac, hsa-miR-548ay-3p, hsa-miR-660-5p, and hsa-miR-6747-3p, did not have any predicted target. Three miRNAs had overrepresented annotations, encompassing almost 75% of the total of predictions; these miRNAs were hsa-miR-210-3p, hsa-miR-1343-3p, and hsa-miR-30b-5p. Conversely, two of them had very few annotations, representing 0.86% (hsa-miR-223-3p) and 0.76% (hsa-miR-3605-3p) of the total number of annotations. More details of the results can be seen in Table 1. The ten most common targets were: ANKRD52 (shared by eight miRNAs), SPEN (seven), MIDN (seven), SRRM2 (six), SON (six), MDM2 (six), LONRF2 (six), LARP1 (six), FBXL19 (six), and DDX3X (six). More details of the results of the target prediction can be seen in Appendix A.

The second step of the functional analysis was a bibliographic search of relevant miRNA targets of the miRNA signature. We found 10 genes (Table 1) directly or indirectly regulated by any of the 16 miRNAs. The main functions implicated were related to the signal transducer and activator of transcription 3 (STAT3) and with several of its repressors (SOCS2, SOCS4, PTPN11). We also found some targets related to the multidrug resistance-associated proteins (ABCC1, ABCB1) and drug metabolism (CYP2C19).

The last step was the network analysis and the functional enrichment of Pathways and GO terms. The most connected sub-network seems to be related to the AP-1 Transcription Factor Subunits (Figure 2). We found an interesting implication of hsa-miR-504-5p, hsa-miR-199a-5p, and hsa-miR-3615 in the regulation of transcription factors recognizing the cAMP response elements (CRE) sequence 5′-TGACGTCA-3′. Hsa-miR-504–5p seems to represses the transcription of “MAF BZIP Transcription Factor G” (MAFG), which directly interacts with the “Basic Leucine Zipper ATF-Like Transcription Factor 3” (BATF3) [35]. BATF3 is known to dimerize with the “Transcription Factor AP-1 Subunit Jun” (JUN) acting as transcriptional repressor of CRE elements [36,37]. On the contrary, hsa-miR-199a-5p has been proved to target JUN and JUNB transcripts, both of which dimerize with BATF3 and “Activating Transcription Factor 4” (ATF4) [38]. Moreover, hsa-miR-3615 directly represses “CREB/ATF BZIP Transcription Factor” (CREBZF) transcript, which is a known repressor of the “CAMP Responsive Element Binding Protein 3” (CREB-3), a transcription factor that binds to the CRE pattern [39]. Finally, the DNA Methyltransferase 1 (DNMT1) and the Sirtuin 1 (SIRT1) are also implied in this subnetwork. SIRT1 deacetylates DNMT1, which is a target of hsa-miR-504-5p [40]. It is worth noting the presence of more miRNA–targets interactions in this subnetwork. These interactions are hidden due to the degree filter applied to declutter the network in order to spot the most connected (and therefore, potentially relevant) nodes.

The top 5 pathways enriched in the Reactome database were, in order of presence in our network: 17.75% of the genes related with the metabolism of proteins (q-value = 5.03^−52^), 14.42% related with the transcription of genes (q-value = 1.6^−51^), 14.12% related to diseases (q-value = 3.89^−42^), 8.04% associated with RNA metabolism (q-value = 1.99^−40^), and 7.48% with a cellular response to stress (q-value = 1.73^−44^). Regarding the WikiPathway database, the top 5 enriched pathways were: 5.51% of genes related to VEGFA-VEGFR2 signaling (q-value = 4.44^−29^), 3.47% related to miRNA effects in Alzheimer’s disease (q-value = 3.6^−19^), 2.39% related to breast cancer (q-value = 1.39^−16^), 1.75% related to the androgen receptor signalling pathway (q-value = 1.94^−15^), and 1.72% related to ionising radiation damage to DNA and the subsequent cellular response via the ATR Serine/Threonine Kinase (q-value = 8.38^−17^). Finally, in the KEGG database, the top 5 enriched pathways were: 3.98% of genes related to amyotrophic lateral sclerosis neurodegenerative disorder (q-value = 4.07^−17^), 3.36% related to Huntington neurodegenerative disease (q-value = 1.41^−14^), 3.28% of the genes related to intestinal infection caused by Shigella (q-value = 3.57^−21^), 2.96% related to Parkinson neurodegenerative disorder (q-value 9.04^−15^), and 2.66% related to viral infection and human carcinogenesis (q-value = 6.45^−17^).

If we look at the genes in these pathways, we can notice that 10 of the 15 enriched pathways have TP53, PSMD4, PSMC1, and MDM2 proteins in common, and 9 pathways also have in common several 26S proteasome non-ATPase regulatory subunits, along with MTOR, EP300, CYCS, and CREBBP.

## 3. Discussion

Schizophrenia is a severe mental disorder that affects around 1% of the world’s population [1]. The current approach for its treatment is antipsychotic medication, but about one-fifth to one-half of patients do not respond to this pharmacological therapy [15]. Patients with resistant schizophrenia have poorer outcomes than patients with other severe mental illnesses, experiencing higher degrees of inadaptation to daily demands [41], which is an indication of a bad prognosis. Here, we present 16 miRNAs that are potentially related to treatment-resistant schizophrenia, and that will be a good starting point to biologically understand this condition and develop biological biomarkers.

### 3.1. miRNAs Previously Associated with Psychiatric Conditions

Seven of these 16 miRNAs have been previously related to neuropsychiatric conditions: **miR-199a-5p** (downregulated in MNR) has been found to target the IFNAR1 gene in a Dual-Luciferase Reporter Assay in a cell-based Parkinson’s disease model [42]. This gene encodes for type I Interferon Receptor 1, whose depletion in the organism seems to be protective against cognitive decline in mouse models of Alzheimer’s disease [43], and its presence in the human entorhinal cortex is proposed as a potential indicator of this condition [44]. Downregulation of this miRNA was also observed in dopaminergic neurons of Parkinson patients [45]. **MiR-296-5p** (downregulated in MNR) was found to be upregulated in schizophrenia patients compared to healthy controls [46]. **MiR-504** (upregulated in MNR) has been found to target the SHANK3 gene in a luciferase assay in cultured mouse hippocampal neurons [47]. This gene encodes for scaffold proteins of the postsynaptic density, and it is also related to synapse formation and dendritic spine maturation; alterations in this gene are related to autism spectrum disorder and schizophrenia type 15 [48]. Moreover, this miRNA was found to be overexpressed in post-mortem brain tissue of patients with bipolar disorder, but not in schizophrenia patients, when compared to healthy controls [49]. **MiR-660-5p** (downregulated in MNR) was upregulated in the serum of patients with Alzheimer disease compared to healthy controls [50], but was found to be underexpressed in Long-Evans rats exposed to ethanol intake when compared to healthy control rats [51]. **MiR-3615** (upregulated in MNR) was included in a 2020 study as one of a seven-miRNA profile for the discrimination of schizophrenia from controls [52]. A downregulation in **miR-30b-5p** levels (downregulated in MNR) was observed in patients with BD in comparison to healthy controls [53]. Finally, **miR-6747-3p** (upregulated in MNR) was found to be upregulated in the serum of Alzheimer’s disease patients compared to healthy controls [54].

### 3.2. miRNAs Potentially Related to Drug Resistance

A total of 3 of these 16 miRNAs have been previously related to the resistance to other medications. **miR-145-5p** (downregulated in MNR) has been linked to the acquired resistance to cisplatin and fluorouracil combination-based chemotherapy in gastric cancer patients [55] and also found to be underexpressed in patients with bipolar disorder type I when compared to healthy controls [56].

Downregulation of **miR-210-3p** (downregulated in MNR) has been related to an increase in ABCC1 expression in renal carcinoma and a subsequent reduction in chemotherapy drug sensitivity. The ABCC1 gene encodes for the multidrug resistance-associated protein 1; a member of the superfamily of ATP-binding cassette (ABC) transporters, it is related to multidrug resistance. Single nucleotide polymorphisms in this gene were associated with alterations in clozapine and norclozapine serum levels [57].

Finally, a 2020 study [29] found an upregulation of **miR-1343-3p** (upregulated in MNR) in the presence of the mutant allele rs4244285 of the CYP2C19 gene, which encodes a member of the cytochrome P450 enzyme superfamily. This rs4244285 allele encodes the CYP2C19*2 variant, which is known to be related to poor metabolism of compounds, such as some antidepressants, including fluvoxamine or sertraline hydrochloride [58,59]; anticonvulsants, such as valproic acid [60]; and benzodiazepines, such as phenazepam [61], among others. Moreover, CYP2C19 product is an enzyme responsible for the metabolization of clozapine [62,63].

### 3.3. Interpretation of the Hierarchical Clustering

In the dendrogram of MR and MNR samples (Figure 1C), a clear separation between the MR and MNR samples can be seen. Some pairs of samples (arrival and departure) are together in the MNR cluster (MR010A/D, MR011A/D, and MR012A/D) and in the MR cluster (MNR018A/D, MNR001A/D, and MNR019A/D). This probably means that the hospital environment did not significantly change their miRNA expression before and after hospitalisation, either due to similar environment variables (e.g., same hours of sleep, meals, light exposure, …) or pharmacological factors (e.g., the same medication is maintained). Interestingly, three pairs of samples (MR014A/D, MR018A/D, and MNR002A/D) change from the MNR cluster to the MR cluster after hospitalisation, whereas in two samples (MNR005A/D and MR017A/D), the opposite phenomena is observed. A change between the MNR cluster to the MR cluster could be the consequence of the hospital’s regular administration of the medication in an adequate dose, contrasting with a previous pharmacological deregularization [64,65]. This explanation is the most feasible, as two of these three samples were from patients treated with multiple antipsychotics, and the third one came from a MR patient treated with clozapine.

On the contrary, a change between the MR cluster to the MNR cluster is more difficult to explain. This implies either a MNR or MR patient was more similar to a MR patient initially, and after hospitalisation, became more similar to an MNR; in other words, a theoretical worsening of the treatment response after hospitalisation.

In the dendrogram of MR, MNR, and PA samples (Figure 1D), it can be observed that the miRNA expression of the TRS profile is more similar between PA samples and MNR samples (7 of 9 PA samples clustered in the MNR cluster, made of 73.9% MNR samples). In other words, MNR samples (treated patients) are more similar to PA (untreated) than to MR (treated). Interestingly, 4 PA samples clustered in the MR cluster, made of 78.9% MR samples. If we look at the clinical records of these four patients, we can observe that P010A and P011A had pathologies unrelated to schizophrenia—the first one, dementia related to a brain tumour and the second one, active consumption of toxic substances and aggressiveness not attributed to a psychiatric pathology. On the contrary, P008A figures as a stable non-medicated patient since the last two years, and P007A had an important mental disability previous to the psychotic episode. All this suggests that the PA samples of the MR cluster probably have conditions not intrinsically related to schizophrenia.

### 3.4. Interpretation of the Functional Analysis

Although at a first glance, the results of the pathway enrichment analysis could seem divergent, most of the pathways have something in common: they are related to the exposure and response of the organism to a stressor. The vulnerability–stress model has been proposed as a broad explanation for the schizophrenia aetiology since the 1980s [66,67], and it is currently used as an explanation of their aetiology and prodromes [68,69,70]. Experimental evidence, such as the association between the level of activation of the inflammatory response and schizophrenia, seems to suggest a role for stress in this condition [71,72,73,74]. Moreover, differences in this inflammatory response have also been noticed between schizophrenia and treatment-resistant schizophrenia [73,75,76,77,78].

If we closely look at the enriched pathways in Table 2, we can notice that most of them have Tumor Protein P53 (TP53) and its regulator MDM2 Proto-Oncogene (MDM2) in common, both heavily involved in the cellular stress response. MDM2 is a primary negative regulatory factor of p53 (the TP53 product) and contributes to its normal expression in the cell. This is achieved by the ubiquitination of p53 by MDM2, which leads to the p53 degradation [79]. On the other side, the protein p53 is a tumour suppressor, which responds to diverse cellular stresses to regulate expression of its target genes; it is involved in the regulation of processes such as cell cycle arrest, apoptosis, senescence, DNA repair, and changes in metabolism. The MDM2/p53 pathway has been found to play an important role in drug resistance [80,81,82,83], and its regulatory expression was found to be altered during treatments with the antipsychotic fluspirilene [84]. Interestingly, the association between the schizophrenia condition and the risk for cancer development has been debated for more than 100 years [85,86,87,88,89,90,91]. The results indicate both a decrease and an increase in the risk of cancer, which varies according to the type of tumour studied and the schizophrenia cohort [90]; still, the evidence shows that the mechanisms that regulate tumorigenesis seem to be closely related to this condition.

### 3.5. A Molecular Model for Treatment Resistant Schizophrenia

In an effort to integrate the results of the DEA, the network and pathway analysis, the miRNA target prediction, and the bibliographic search, we propose in Figure 3 a pathway in which the DE miRNAs could be potentially implied. Hopefully, it could bring a hint about the treatment-resistant schizophrenia aetiology.

We found that p53/MDM2 interaction could possibly be the key to explain the observed differences in miRNA expression between the MR and MNR groups. First, both molecules were the most recurrent similarity in the top five enriched pathways in three different databases. Second, MDM2 and its regulators, MAGE Family Member C2 (MAGEC2) and (RING Finger Protein 96) TRIM28, were targeted by eight different miRNAs of the TRS profile. Third, SIRT1 and its deacetylation substrate, DNMT1, are both present in the most connected subnetwork and have a direct involvement in the regulation of p53 transcription. Finally, bibliographic evidence supports the proposed pathway in four aspects: (i) the implication of p53/MDM2 in the multidrug resistance process [80,81,82,83], (ii) the involvement of p53 in autophagy, a process known to be related to antipsychotic response [92,93,94,95]; (iii) the observed dysregulations of this pathway in schizophrenia patients [96,97,98], and (iv) the relationship of this pathway to negative symptoms, such as memory, learning ability impairment [99], and major depression [100].

In this pathway, SIRT1 deacetylates the methyltransferase DNMT1. This results in an activation or repression of its methyltransferase activity, which varies with the deacetylation position [40]. For its part, increased levels of DNMT1 seem to enhance the methylation in the TP53 promoter, leading to the downregulation of p53 [101,102,103]. Moreover, SIRT1 has also been found to directly deacetylate p53, repressing its transactivation [104,105,106], and play a role in the modulation of the response to stress. Two recent examples of this are the observed increment in levels of p53 and decreased levels of SIRT1 in mice brain after long-term “ultraviolet A” eye radiation [99], as well as the inverse phenomena after acute heat stress in bovine granulosa cells, where decreased levels of p53 and increased levels of SIRT1 were observed [107]. In MNR, SIRT1 is potentially targeted by the underexpressed hsa-miR-30b-5p, whereas DNMT1 is by the overexpressed hsa-miR-504-5p (Figure 3).

Other regulators of p53 are MDM2 and MAGEC2. The MDM2 transcript is heavily targeted by the miRNAs of our TRS profile, and its protein product is well-known for its role in p53 inactivation [83,108,109]. Under nonstressed conditions, MDM2 regulates p53 expression through an autoregulatory feedback loop [110,111,112]. This regulation can be made by the direct binding of MDM2 to p53 transactivation domain, by MDM2 involvement in the nuclear translocation of p53, or by promoting its degradation using its ubiquitin ligase activity [108,113]. MDM2/p53 regulation has been extensively linked to drug resistance [80,81,114], and there is evidence of its regulatory interaction being disrupted by some antipsychotics, such as fluspirilene and pimozide, or mood stabilisers, such as lithium [84,115,116]. In MNR, MDM2 is potentially targeted by seven miRNAs of the TRS profile (Figure 3), four downregulated (hsa-miR-3127-3p, hsa-miR-210-3p, hsa-miR-145-5p, and hsa-miR-30b-5p) and three overexpressed (hsa-miR-1343-3p, hsa-miR-3127-3p, and hsa-miR-504-5p).

Regarding MDM2 regulators, an important modulator of MDM2 ubiquitin ligase activity is MAGEC2, which has recently been found to inhibit the ubiquitination of p53 by directly interacting with the MHD domain of MDM2 [117]. At the same time, TRIM28 competes with MDM2 for MAGEC2 binding, thus acting as a promoter of MDM2 ubiquitin ligase activity and, consequently, triggering p53 degradation [117,118]. In MNR, TRIM28 is potentially targeted by four miRNAs of the TRS profile (Figure 3), two overexpressed (hsa-miR-3605-3p, hsa-miR-1343-3p) and two downregulated (hsa-miR-296-5p, hsa-miR-145-5p).

Finally, the acetylation of p53 is enhanced as a stress response, which results in its transcriptional activation [119,120,121]. The activated p53 acts as a transcription factor, promoting the expression of several genes, from which we selected Tumor Protein P53 Inducible Nuclear Protein 1 (TP53INP1) as the most relevant in TRS for two reasons: (i) it has been the only one targeted by two of the miRNAs of the TRS profile (Figure 3) and (ii) the known interactions of its product and several GABA Type A Receptor-Associated Proteins (GABARAP, GABARAPL1/L2) [122,123,124], which are known to mediate GABA-A receptor intracellular transport [125,126]. The availability of these GABA-A receptors has been related to the severity of symptoms in schizophrenia [127], and elevated levels of its ligand GABA had been reported in the midcingulate cortex in TRS patients [128]. In MNR, TP53INP1 is potentially targeted by two miRNAs of the TRS profile (Figure 3), one overexpressed (hsa-miR-504-5p) and one downregulated (hsa-miR-30b-5p).

Although there is not much research about the role of p53 in TRS, there is some evidence linking alterations of p53 expression, its activation, and polymorphisms in TP53 with higher schizophrenia risk and symptoms severity [97,98,129,130,131,132]. Decreased levels of SIRT1 in plasma has been linked with a higher comorbidity of depressive symptoms [133,134,135], whereas increased levels of DNMT1 had been observed in the brains of schizophrenia and bipolar patients [136,137,138,139]. Finally, MDM2 had been found to be underexpressed in the dorsolateral prefrontal cortex of schizophrenia patients [96].

All this together points to an atypical response to stress after antipsychotic administration as one key factor in the development of antipsychotic resistance. Our results indicate that this response could be potentially mediated by p53, with an important implication of its regulators MDM2, TRIM28, SIRT1, and DNMT1. Future research on TRS might focus on comparing the expression of these molecules between TRS and non-TRS patients.

### 3.6. Limitations and Future Perspectives

This study has the following limitations. (i) Despite the fact that we defined a TRS fingerprint based on the miRNA differential expression between MR and MNR groups, the mechanism through which those miRNAs affect the resistance condition is still unknown. (ii) Whether peripheral miRNAs represent changes in the central nervous system was not assessed. Further experiments analysing cerebral spinal fluid in human samples might be an interesting approach. (iii) Although we corrected for processing batch; sex; drug consumption; time; treatment based on the subgroup of the diazepines, oxazepines, thiazepines, and oxepins; and treatment based on other antipsychotics, it could have been of interest to also correct for treatments based on clozapine and the presence of cognitive decline. (iv) It might have been interesting to study the miRNA response to specific antipsychotic therapies. (v) Results of the hierarchical clustering suggest that the TRS profile is partially associated with the schizophrenia condition, not only with antipsychotic resistance. (vi) The sample size is modest; therefore, it is difficult to predict the scope of our conclusions. It would be necessary to assess the expression of the TRS profile in an independent sample of patients. (vii) At the time of writing this paper, four miRNAs, namely, hsa-miR-548ac, hsa-miR-548ay-3p, hsa-miR-660-5p, and hsa-miR-6747-3p, did not have any predicted target in the last version of Tarbase v.8. Therefore, our functional analysis and the conclusions drawn from it are based on 12 of the 16 miRNAs of the TRS profile. It would be interesting to repeat the functional analysis in the near future.

## 4. Materials and Methods

### 4.1. Experimental Design

We recruited schizophrenia patients with a normal antipsychotic response (MR; *n* = 19), schizophrenia patients with antipsychotic resistance (MNR; *n* = 21), patients with a first psychotic episode (P; *n* = 13), and 43 healthy individuals (C; *n* = 43). Two blood samples were collected per patient, the first one at the hospital arrival (MRA, MNRA, PA) and the second one at the hospital discharge (MRD, MNRD, PD). Scores on PANSS (positive, negative, general) and SAAS scales were recorded at the arrival [140,141]. P patients were followed up until they could be classified as responders or nonresponders to medication, resulting in eight PA with normal antipsychotic response (PA_R_ = 9) and four PA with antipsychotic resistance (PA_NR_ = 4). As treatment resistance is a chronic condition in schizophrenia, and environmental changes during hospital admission can disturb miRNA expression, we will consider arrival and discharge samples from the same patient as two replicas of the same condition. Thus, differences between the MR and MNR groups will be more likely to be related with TRS and not with environmental variations. From now on, we will use MR to refer to MRA and MRD samples and MNR to refer to MNRA and MNRD samples.

To find a miRNA signature of the antipsychotic response, we looked for differences in miRNA expression between the MR and MNR groups. The resulting “miRNA signature” was used to perform a hierarchical clustering with the MR, MNR, and PA samples; we are interested in the clusters assigned to PA samples, as well as the overall distribution of groups between clusters. As a first approach to outline the biological context of these results, we performed a target prediction, followed by a network and a functional analysis of the miRNAs of the signature.

### 4.2. Human Participants Included in the Study

Patients were recruited from Álvaro Cunqueiro Hospital (Vigo, Spain) between September 2018 and November 2021. Inclusion criteria were: (i) meeting the DSM-V schizophrenia diagnostic criteria, (ii) aged 18 years or older, and (iii) signed written consent. The exclusion criteria were an additional DSM-V diagnosis, medical illnesses, and neurological conditions, as well as pregnancy or lactation. Age, sex, drug consumption (alcohol/tobacco/illegal), and current pharmacological treatment were recorded for each patient (Table 3).

All patients and controls for this study were of Spanish nationality. The research was conducted in accordance with the Declaration of Helsinki, and all relevant ethical approvals were obtained. Written consent was obtained from all patients or their corresponding legal guardians.

### 4.3. Blood Collection

Two blood samples per patient were drawn from the cubital vein and collected into PAXgene tubes (BD Biosciences, Franklin Lakes, NJ, USA) in the morning between 8 and 10 a.m, the first one at the hospital arrival and the second one at the hospital discharge. Samples were incubated 2 h at room temperature to ensure complete lysis of blood cells. PAXgene tubes were stored in the freezer (−80 °C) following commercial specifications (Qiagen, Hilden, Germany). The tubes were first transferred to −20 °C (24–72 h) and then transferred to −80 °C freezer until further processing.

### 4.4. Total RNA Purification from Human Blood Samples

Total RNA, including small RNA, was purified from whole blood samples using the PAXgene^®^ Blood microRNA Kit from Qiagen (PreAnalytiX GmbH, Hombrechtikon, Switzerland). PAXgene Blood microRNA Kit isolates total RNA >18 nucleotides (including miRNA) from human whole blood. Before isolation, samples were put at room temperature overnight. We used Automated purification of total RNA, including miRNA, on QIAcube instruments, following the manual specification (Qiagen, Hilden, Germany). Briefly, tubes were centrifuged for 10 min at 4000× *g*, and supernatant was removed. A total of 4 mL of RNAse-free water was added to the pellet and then vortexed until the pellet was visibly dissolved. After that, ten minutes of centrifugation at 4000× *g* were applied. This step was repeated once more. After that, pellet was resuspended in buffer BM1, transferred to the 2 mL processing tube, and loaded into the QIAcube Connect shaker. Proteinase K and Buffer BM2 were added prior to the 10 min period of incubation at 55 °C. Incubated tubes were transferred to PAXgene Shredder Spin Column, where 3 min of centrifugation at full speed was applied. Supernatant was transferred into new tubes, and isopropanol was added. Samples were loaded on PAXgene RNA spin columns that bind total RNA >18 nucleotides (including miRNA). Centrifugation at 10,000× *g* for 1 min was applied. Samples were washed once for 15 s with buffer BM3 and centrifuged at 10,000× *g* for 1 min. A DNA digestion was performed by incubating samples with DNase solution for 15 min at room temperature, followed by a centrifugation at 10,000× *g* for 1 min. Spin columns were washed, first with Buffer BM3 and then with Buffer BM4, at 10,000× *g* for 2 min each. Samples were eluted with 50 µL of BR5 Buffer to microcentrifuge tubes. One minute of centrifugation at 10,000× *g* was applied prior to the sample incubation at 65 °C for 5 min in the QIAcube Connect shaker. Finally, samples were immediately transferred to ice after incubation and then stored at −80 °C until further use.

### 4.5. Small RNA Sequencing

For RNA quality control, a Bioanalyzer Instrument (Agilent Technologies, Santa Clara, CA, USA) was used. For concentration determination, a NanoDrop (Thermo Scientific, Waltham, MA, USA) was used. In addition, for evaluating RNA sample quality prior to miRNA/small RNA NGS library preparation, a QIAseq miRNA Library QC Spike-ins kit was used (Qiagen, Hilden, Germany). Library preparation was done using NEBNext Multiplex Small RNA Sample Prep Set for illumina (New England biolabs, Ipswich, MA, USA) to produce high quality microRNAome data from human blood samples. Briefly, the total amount of 150 ng of RNA was taken for further cDNA preparation, fragmentation, adapter ligation, and hybridization. After pooling the libraries together, polyacrylamide gel electrophoresis was run for size selection. The insert size of 150 base pairs (bp) was chosen for quantification and purification. Quality was checked on Bioanalyzer Instrument, and libraries were quantified using a Qubit 2.0 Fluorometer (Life Technologies, Carlsbad, CA, USA). Sequencing of 8 pM concentration was performed on HiSeq 2000 sequencing system (Illumina, San Diego, CA, USA) using a 24 samples per Lane, 50 bp single read setup. As a control for Illumina sequencing runs, a PhiX Control that is a reliable adapter-ligated library was used. Sequencing data were demultiplexed using CASAVA v1.8 (Illumina, San Diego, CA, USA), and raw fastq files were generated.

### 4.6. Bioinformatics Analysis

The analytical methodology is detailed in Appendix A. For quality control of the raw data, adapter removal, alignment, and quantification, we used the software myBrain-Seq v0.1.0 [23]. The biological references used were the GRCh38 human genomic build [142] for the alignment and miRBase v.22 database [143] for the miRNA annotation. We filtered miRNAs with less than 10 counts in total using a custom R script. Differential expression analyses were performed using DESeq2 v1.32.0 [144], adapting the script of myBrain-Seq to adjust for six potential confounding variables: (i) processing batch, (ii) sex, (iii) drug consumption (alcohol OR tobacco OR illegal), (iv) time (arrival, discharge), (v) treatment based in the subgroup of the diazepines, oxazepines, thiazepines, and oxepins, and (vi) treatment based in other antipsychotics. To obtain a miRNA signature for the resistance to medication, we first performed the DEA comparing MR vs. MNR, then subtracted from this profile the common miRNAs between DEA comparisons MNR vs. C and MR vs. C. This was done to filter miRNAs with potentially similar roles in MR and MNR conditions and to obtain the “miRNA profile” (Appendix A, point 3).

### 4.7. Hierarchical Clustering

We performed two different hierarchical clustering analyses using the miRNAs found in the “miRNA profile”: the first one with MR and MNR samples, the second one with MR, MNR, and PA samples. Prior to the hierarchical clustering, the miRNA counts of the profiles were normalised using the Relative Log Expression method of DESeq2 1.32.0 [144]. We used the R package hclust [24] for clustering, using the Euclidean distance metric and Ward method (ward.D2).

### 4.8. Functional Analysis

To narrow down the potential implication of the differentially expressed miRNAs on the antipsychotic resistance, we conducted a target prediction analysis, bibliographic search, and network analysis. For the prediction of the miRNA targets, we used TarBase v8 [27] and a custom bash script. The bibliographic search was performed in order to add documented targets not present in TarBase v8; we looked for genes related to chemoresistance or the metabolism of antipsychotics, or those involved in the diffusion of molecules through the blood–brain barrier. To perform the network analysis, we used Cytoscape 3.9.1 [25]. We started by importing the miRNA–gene interactions obtained from TarBase v8 and the literature search. Then, we selected the miRNAs and ran the Cytoscape default heat diffusion algorithm [145]. Selected nodes were used as a query in the Uniprot database [146], and the resulting networks were merged with the miRNA–gene network. Finally, we used StringApp 2.0.0 [26] to annotate proteins, to filter by tissue, and to perform the functional enrichment analysis of pathways. We kept proteins expressed only in the category of “nervous system” with a tissue score of 4.50 and in the category of “blood” with a tissue score of 3.00. For the pathway enrichment analysis, we used Reactome Pathways [147], WikiPathways [148], and KEGG [62] Pathways databases. To find highly connected subnetworks, we applied a degree filter of value 10.

## 5. Conclusions

We presented a 16-miRNA profile related to treatment-resistant schizophrenia. With this profile, we performed a hierarchical clustering of MR and MNR samples that resulted in two clusters: the first one with 73.2% of the MR samples and the second one with 72.4% of the MNR samples. We did not use the PANSS and SAAS scores to improve the clustering accuracy, as we did not find associations between PANSS and SAAS scores and the TRS condition. By observing the dendrogram of MR, MNR, and PA samples, we deduced that similarities between the TRS profile of patients with a first psychotic episode and schizophrenia patients’ response to medication seem to be potential evidence of a first psychotic episode unrelated to schizophrenia. The network and functional analysis of the TRS profile suggests that atypical response to stress after antipsychotic administration could be one key factor in the development of antipsychotic resistance. Finally, we condensed all the results in a molecular pathway that is presumably altered in TRS patients, with a pivotal role of p53 and its regulators MDM2, TRIM28, SIRT1, DNMT1, and eight miRNAs of the TRS profile.

## Figures and Tables

**Figure 1 ijms-24-01891-f001:**
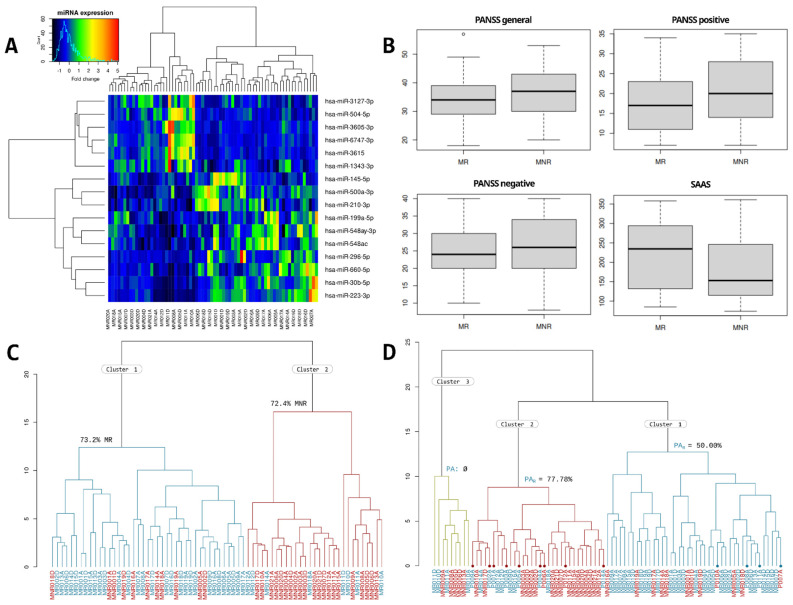
(**A**) Heatmap with the results of the hierarchical clustering of MR and MNR samples. (**B**) Box-plots with the MR and MNR punctuations on the PANSS general, positive, negative, and SAAS scales. The overlap between both groups can be seen. (**C**) Hierarchical clustering with MR samples (blue) and MNR samples (red); more than 70% of samples of both conditions were clustered homogeneously. (**D**) Hierarchical clustering with samples in MR, MNR, and PA groups. MR samples are colored in blue, MNR samples in red and PA samples were marked with a dot and colored in blue or red depending on the evolution of their psychotic episode.

**Figure 2 ijms-24-01891-f002:**
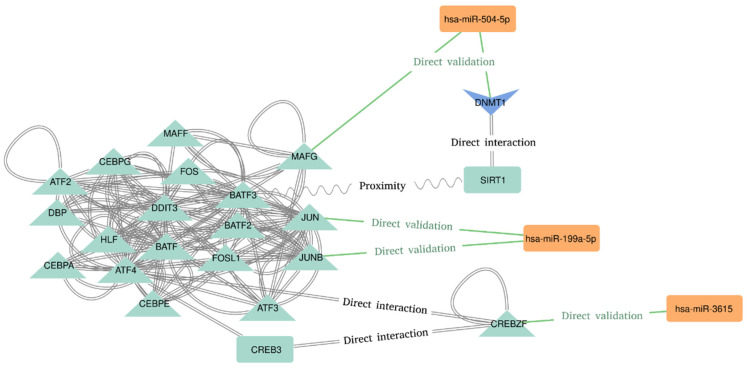
Molecular interactions deduced from the most connected nodes of the network.

**Figure 3 ijms-24-01891-f003:**
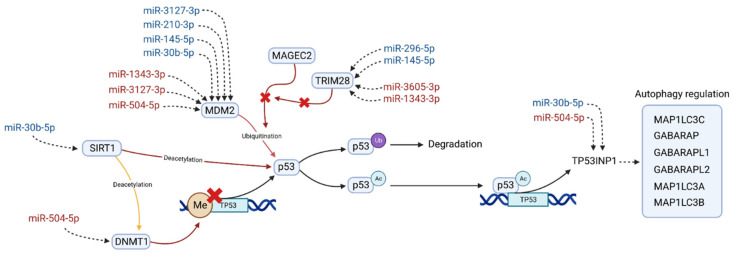
Pathway in which differentially expressed miRNAs could be involved, potentially related to the antipsychotic resistance in schizophrenia. MiRNAs are colored according to their expression in the MNR group when compared to the MR group; red indicates overexpression and blue underexpression. Sirtuin 1 (SIRT1) interacts with the DNA Methyltransferase 1 (DNMT1) and the Tumor Protein P53 (p53) regulating its expression. Proteins MDM2 Proto-Oncogene (MDM2), MAGE Family Member C2 (MAGEC2), and RING Finger Protein 96 (TRIM28) are closely linked with p53 inactivation and heavily targeted by the miRNAs of our TRS profile. This inactivation of p53 is through ubiquitination (p53^Ub^), which leads to its degradation. Acetylated p53 (p53^Ac^) acts as a transcriptional factor of several genes, from which Tumor Protein P53 Inducible Nuclear Protein 1 (TP53INP1) is the one most targeted by the TRS profile. This protein is a positive regulator of the autophagy and also acts as a transcription regulator in response to cellular stress. This regulation is made through its interaction with GABA Type A Receptor-Associated Protein (GABARAP), with GABA Type A Receptor Associated Protein Like 1/2 (GABARAPL1, GABARAPL2) and with Microtubule Associated Protein 1 Light Chain 3 Alpha/Beta/Gamma (MAP1LC3A, MAP1LC3B, MAP1LC3B).

**Table 1 ijms-24-01891-t001:** Number of targets per miRNA of the TRS profile returned by TarBase v.8 (Targets in TarBase) and found in the literature (Literature targets).

Name	Targets in TarBase	Literature Targets	Literature Reference
hsa-miR-210-3p	4075	ABCC1	[28]
hsa-miR-1343-3p	3148	CYP2C19	[29]
hsa-miR-30b-5p	2688	SOCS1, SOCS3	[30]
hsa-miR-145-5p	940	-	-
hsa-miR-296-5p	547	-	-
hsa-miR-199a-5p	396	STAT3	[31]
hsa-miR-500a-3p	371	SOCS2, SOCS4, PTPN11	[32]
hsa-miR-3127-3p	354	RAP2A	[33]
hsa-miR-3615	274	-	-
hsa-miR-504-5p	263	-	-
hsa-miR-223-3p	114	ABCB1	[34]
hsa-miR-3605-3p	97	-	-
hsa-miR-548ac	0	-	-
hsa-miR-548ay-3p	0	-	-
hsa-miR-660-5p	0	-	-
hsa-miR-6747-3p	0	-	-

**Table 2 ijms-24-01891-t002:** Top 5 enriched pathways per database (Reactome Pathways, WikiPathways, and KEGG Pathways). The column “Gene” is the number of genes in the network associated with each pathway term. The column “Proportion” is the ratio between the number of genes with that term divided by the total number of genes in the network (*n =* 3718).

Database	Pathway	Genes	FDR Value	Proportion
Reactome Pathways	Metabolism of proteins	660	5.03 × 10^−52^	17.75%
Reactome Pathways	Gene expression (Transcription)	536	1.6 × 10^−51^	14.42%
Reactome Pathways	Cellular responses to stress	278	1.73 × 10^−44^	7.48%
Reactome Pathways	Disease	525	3.89 × 10^−42^	14.12%
Reactome Pathways	Metabolism of RNA	299	1.99 × 10^−40^	8.04%
WikiPathways	VEGFA-VEGFR2 signaling pathway	205	4.44 × 10^−29^	5.51%
WikiPathways	Alzheimer’s disease and miRNA effects	129	3.6 × 10^−19^	3.47%
WikiPathways	DNA IR-damage and cellular response via ATR	64	8.38 × 10^−17^	1.72%
WikiPathways	Integrated breast cancer pathway	89	1.39 × 10^−16^	2.39%
WikiPathways	Androgen receptor signaling pathway	65	1.94 × 10^−15^	1.75%
KEGG Pathways	Shigellosis	122	3.57 × 10^−21^	3.28%
KEGG Pathways	Amyotrophic lateral sclerosis	148	4.07 × 10^−17^	3.98%
KEGG Pathways	Viral carcinogenesis	99	6.45 × 10^−17^	2.66%
KEGG Pathways	Parkinson disease	110	9.04 × 10^−15^	2.96%
KEGG Pathways	Huntington disease	125	1.41 × 10^−14^	3.36%

**Table 3 ijms-24-01891-t003:** Summary of the samples included in this study.

Phenotype	Males	n	$\overset{-}{x}$ Age	Alcohol	Tobacco	Illegal
P	69.23%	13	32.1	6	6	6
MR	73.68%	19	40.0	5	11	12
MNR	61.90%	21	39.7	7	12	7
C	55.81%	43	41.5	0	4	0
Total		96	39.5	18	33	25

## Data Availability

Data accessible at NCBI GEO database, accession GSE223043 (https://www.ncbi.nlm.nih.gov/geo/query/acc.cgi?acc=GSE223043).

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
