# Peer review of "MiRNA Differences Related to Treatment-Resistant Schizophrenia"

_ijms, 2023, doi:10.3390/ijms24031891_

Round 1

Reviewer 1 Report

In the manuscript titled “ MiRNA differences related to treatment-resistant schizophrenia”, the authors attempted to characterize a miRNA signature for treatment-resistant schizophrenia using schizophrenia patient samples. They have proposed a panel of 16 miRNAs related to treatment resistance schizophrenia and bioinformatically performed network and functional analysis of the miRNAs and their predicted targets. While the manuscript presented a quite comprehensive overview of the miRNA profile and proposed possible mechanisms underlying treatment-resistant schizophrenia, I think the following points have to be addressed prior to publication.

Major comments

1. One of my major concerns about the results is a lack of validation of their miRNAome results. While next-generation sequencing techniques are powerful in detecting differentially expressed genes between samples, a more targeted validation (e.g. using qRT-PCR) is often required. Also, it would be informative to present some basics of the dataset, e.g. sequencing coverage and depth, to gauge the quality of the library and sequencing results.

2. Another major comment pertains to the Results sections. While most text accurately describes the results in the figures, I would suggest including the authors’ interpretation and conclusion at the end of each result section. This would significantly help the flow of the manuscript.

3. Result 3.3. Again, I think validating at least some of the predicted targets would be required to substantiate the authors' argument unless certain targets have been previously established and validated, in which case reference should be included. The authors, however, seemed to have little interest in following up on some of the very promising targets that are shared by half of the proposed miRNA signature panel.

4. Reference to literature is missing. For example, lines 268-270, lines 271-272, lines 272-275, and line 276, etc. Please check through the manuscript to make sure references are included where needed.

Minor comments:

1. Line 222 missing word, “the of the TRS profile”

Author Response

Reviewer 1

Dear Reviewer

First of all, thank you very much for your comments, we will try to satisfy your demands whenever possible.

Reviewer 1, point 1:

“One of my major concerns about the results is a lack of validation of their miRNAome results. While next-generation sequencing techniques are powerful in detecting differentially expressed genes between samples, a more targeted validation (e.g. using qRT-PCR) is often required. Also, it would be informative to present some basics of the dataset, e.g. sequencing coverage and depth, to gauge the quality of the library and sequencing results.”

We appreciate this comment and understand the reviewer's concerns. The sequencing depth of the samples (~34M on average) far exceeds the minimum value range recommended for miRNA-Seq data used for differential expression analysis (~1-2M) (Campbell et al., 2015, 10.1261/rna.046060.114; Metpally et al., 2013, 10.3389/fgene.2013.00020). Moreover, despite the extended use of qRT-PCR, we think it has some drawbacks to consider when applied to a study like this, where the aim is to look for miRNA differences between two conditions. These limitations are well stated in Campbell et al., 2015: (i) qPCR is not as sensitive as sequencing, so subtle but consistent differences in expression can be overlooked by qPCR, limiting our vision of the real molecular scenario. (ii) qPCR and sequencing are very different technologies with their own biases, thus it is difficult to compare its results directly. We strongly believe that our study can contribute to the understanding of the treatment-resistance schizophrenia condition, as other publications did without the need of experimental validation, some examples of these publication are: Hoss et al., 2016 (10.3389/fnagi.2016.00036); Nie et al., 2020 (10.3389/fnins.2020.00438); Pala & Denkçeken, 2020 (10.1089/cmb.2019.0412).

We also thank the reviewer for suggesting adding more information about the dataset such as sequencing coverage and depth, we totally agree that these data will help to improve the study. However, as we are handling transcriptomic (miRNA) data, we believe that the sequencing coverage is not representative of the data quality. This is because the reads will align only with the loci that transcribe miRNAs and not with the entire genome, resulting in very low coverage values. In this case, the second parameter proposed by the reviewer, sequencing depth, is more adequate. Sequence depth is the recommended measure for data quality of an RNA-Seq experiment (Liu et al., 2014, 10.1093/bioinformatics/btt688). We added the Supplementary Table 1 (and its mention in section 3.1) with the sequencing depth of the samples and the proportion of assigned and mapped sequences as measures of quality for our dataset.

Reviewer 1, point 2:

“Another major comment pertains to the Results sections. While most text accurately describes the results in the figures, I would suggest including the authors’ interpretation and conclusion at the end of each result section. This would significantly help the flow of the manuscript.”

We thank the reviewer for this suggestion. We understand from this recommendation that the interpretation of the results and the conclusions that are present in the Discussion section should be moved to the corresponding section of Results. After carefully considering this interesting approach, we believe that the current structure of the Results and Discussion section is the optimal for efficiently transmitting the information. We believe that keeping both sections separate facilitates the readability of the results and avoids confusion between empirical data and interpretations. This could be especially important in this article given the variety of results and the breadth of the discussions. In addition, the interpretation of the results is used as a starting point to deepen the discussions. By separating them, either the same information is repeated in both sections, or the reasoning linking the results with the interpretations is interrupted.

Reviewer 1, point 3:

“Result 3.3. Again, I think validating at least some of the predicted targets would be required to substantiate the authors' argument unless certain targets have been previously established and validated, in which case reference should be included. The authors, however, seemed to have little interest in following up on some of the very promising targets that are shared by half of the proposed miRNA signature panel.”

We understand these concerns and apologize for any confusion. Targets were not predicted by an algorithm but annotated using a database of experimentally supported miRNA-gene interactions (Tarbase v.8). This implies that all the miRNA targets we used in the functional analysis were experimentally validated (usually by more than a study) and can be easily traced back usingt this public database. We added the Supplementary Table 2 with the results of TarBase v.8, where more details regarding the validation of the miRNA-gene interaction are given (cell line, tissue, technology, validation type, regulation…). We mentioned this supplementary table in section 3.3, in line 253.

On the other hand, the number of annotations for a given miRNA will depend on the previous knowledge about that miRNA, which means that a bias in the number of targets (and coincidences) is expected. It is briefly indicated in section 3.3 (lines 246-250) “Three miRNAs had overrepresented annotations, encompassing almost 75% of the total of predictions [...]” and the bias shown in Table 2 (column Targets in TarBase). That bias is the main reason behind not focusing our interest on the most shared targets alone and to consider a broader approach as an optimum solution for suggesting relevant targets (network analysis, bibliographic search of potential targets, enriched pathways).

Reviewer 1, point 4:

“Reference to literature is missing. For example, lines 268-270, lines 271-272, lines 272-275, and line 276, etc. Please check through the manuscript to make sure references are included where needed.”

We thank the reviewer for detecting this issue, we added the missing citations in section 3.3 accordingly (references 35-40) and revised the full manuscript to ensure that all references were included where needed.

Reviewer 1, point 5:

“Line 222 missing word, “the of the TRS profile””

We added the missing word: “the discriminative power of the TRS profile”.

Again, thank you very much for improving the exposition of our work with your thoughts.

Sincerely yours,

Roberto Agís, Hugo López, and Daniel Pérez

Reviewer 2 Report

Present study related to Treatment resistant schizophrenia (TRS) is very interesting and well designed. On the other hand, there are some concerns to be solved before publication in International Journal of Molecular Sciences.

1-     How authors decide the numbers of samples are statistically sufficient in this study? Did they use a statistical calculation like power analysis?

2-     I couldn’t find the ethical committee permission file number. Committee title, approval date and permission of ethics document's number must be stated.

3-     What was the role of genders in this study? Was there any difference between male and female patients?

4-     Is there any mechanistical relationship between cancer and dementia related genes and schizophrenia pathogenesis.

5-     Pathway analyses were seemed too general. Related genes could be discussed in a specific manners like schizophrenia mechanism/progression relationship.

Some recommendations.

Line 128 “Add 4 ml RNAse-free water to the pellet”, may change into “4 ml of RNAse-free water was added to the pellet”. similar phrase may be fixed.

Author Response

Reviewer 2

Dear Reviewer

First of all, thank you very much for your comments, we will try to satisfy your demands whenever possible.

Reviewer 2, point 1:

“How authors decide the numbers of samples are statistically sufficient in this study? Did they use a statistical calculation like power analysis?”

The data from this study steamed from a project whose objective is the study of schizophrenia as a general condition. On that project, sample size estimation was of N = 100 per group (C and SZ patients). However, sample collection was constrained by many factors such as limited resources or unlikely events (such as covid-19 pandemic). Moreover, the process of recruitment took us almost 3 years with an in-between pandemic which ultimately complicated the process. In this study, our intention was to subdivide the SZ group into responders to medication (MR = 19), non-responders to medication (MNR = 21), patients with a first episode (P = 13) and healthy controls (C = 43). After discussing with our support team of statistics, we decided to conduct the study with this sample size, as it is similar to the commonly found in other studies of the same nature. For example: Hoss et al., 2016 (10.3389/fnagi.2016.00036) recruited 29 vs 33 (Parkinson’s disease vs control); Hicks et al., 2016 (10.1186/s12887-016-0586-x) recruited 24 vs 21 (Autism spectrum disorder vs control); Ma et al., 2018 (10.1016/j.psychres.2018.03.080) recruited 10 vs 10 (first-onset schizophrenia patients vs control); Nie et al., 2020 (10.3389/fnins.2020.00438) recruited  5 vs 34 (Alzheimer’s disease vs control) and 7 vs 34 (Parkinson’s disease vs control) or Wang et al., 2018 (10.3389/fpsyt.2018.00227) that recruited 5 vs 5 (ADHD vs control). We also advise about the limitations in the scope of our conclusions due the sample size (section 4.6, lines 524-526).

Reviewer 2, point 2:

“I couldn’t find the ethical committee permission file number. Committee title, approval date and permission of ethics document's number must be stated.”

We thank the reviewer for spotting this issue, we added the information in the “Institutional Review Board Statement” section: “(protocol code 2016/577, 2 March 2017)”.

Reviewer 2, point 3:

“What was the role of genders in this study? Was there any difference between male and female patients?”

We thank the reviewer for this question. The proportion of males and female patients is summarized in table 1. The statistical model used for differential expression analysis was also corrected by “sex”, to avoid the influence of this factor in the results.

Reviewer 2, point 4:

“Is there any mechanistical relationship between cancer and dementia related genes and schizophrenia pathogenesis?”

We appreciate this interesting question. To the best of our knowledge, there are no mechanistic relationships at this time relating these three concepts (cancer and dementia genes, schizophrenia pathogenesis). However, there seem to be some association between cancer incidence and schizophrenia (discussed in section 4.4), a strong association between schizophrenia and depression (Buckley et al., 2009, 10.1093/schbul/sbn135), and a significant association between non-affective psychotic disorders and a higher risk of dementia (Miniawi et al., 2022, 10.1017/S0033291722002781). Nonetheless, we think deepening these associations are beyond the scope of our study, and that this interesting question should be addressed in future research.

Reviewer 2, point 5:

“Pathway analyses were seemed too general. Related genes could be discussed in a specific manners like schizophrenia mechanism/progression relationship”.

We are grateful for this interesting comment, we would like to develop the reasoning that led us to give such orientation to the discussion of the pathway analysis. Although the objective of the study is to analyze the differences in miRNA expression between patients with schizophrenia and patients resistant to schizophrenia, it is necessary to consider that the condition "resistance to medication" could be independent of the “schizophrenia” condition. Therefore, the focus of the article has always been to impartially discuss the results without assuming, at any time, the existence of such association. In our opinion, such a strategy avoids a biased interpretation and allows for a rigorous integration of our results in the current knowledge. This is the reason why the results of the pathway analysis are not focused on their connection with schizophrenia, but rather on their relationship with possible mechanisms associated with drug resistance, relating them to schizophrenia when we consider that such relationship was justified.

Reviewer 2, point 6:

“Some recommendations. Line 128 “Add 4 ml RNAse-free water to the pellet”, may change into “4 ml of RNAse-free water was added to the pellet”. similar phrase may be fixed.”

We thank the reviewer for spotting this issue, we changed the text accordingly.

Again, thank you very much for improving the exposition of our work with your thoughts.

Sincerely yours,

Roberto Agís, Hugo López, and Daniel Pérez

Reviewer 3 Report

1.     Keywords are missing.

2.     Introduction is insufficient and references in introduction need to be enriched with other recent articles.

3.     It is necessary to explain what is the originality and novelty of this study in comparison with other studies in the literature.

4.     The presentation and the objective of the study are too long in introduction (l.61-75). Moreover, results should not appear in the introduction, it looks like an abstract.

5.     The authors must put the reference numbers instead of the authors' names in table 2.

6.     The figures (1, 2, 3) should be modified to be more visible and axes must be specified.

7.     Some discussion parts need to be modified. Be careful to distinguish between the results and discussion parts. In sub-section 4.3, a discussion related to previous studies is missing. Titles 4.3 and 4.4 should be changed. The molecular model (4.5) could be integrated into the results.

8.     The authors have to check the references, some have to be redone or completed (i.e. 8, 20).

Author Response

Reviewer 3

Dear Reviewer

First of all, thank you very much for your comments, we will try to satisfy your demands whenever possible.

Reviewer 3, point 1:

“Keywords are missing”.

We appreciate this remark, we have now added the keywords.

Reviewer 3, point 2:

Introduction is insufficient and references in introduction need to be enriched with other recent articles.

We appreciate this reviewer's remark. However, we think that the current introduction is enough to introduce the topics addressed in our study:

  1. Introduction to schizophrenia (lines 42-44).
  2. introduction to treatment resistance schizophrenia (lines 44-54)
  3. introduction to the miRNAs (lines 54-58)
  4. introduction to the problem (lines 59-63)
  5. introduction of what we have done to “solve” the problem (lines 63-71)
  6. introduction to the aim of the study (lines 72-73)
  7. brief summarization of the results (lines 73-77).

We would appreciate it if the reviewer could give us a little more detail on which of the above sections they think should be expanded, or which new section should be added.

Regarding the citations, we made the bibliography search in PubMed and all of the references that we included are still being referenced in present articles. Following the reviewer recommendations, we added the following citations:

  1. (Potkin et al., 2020):  line 58.
  2. (Nucifora et al., 2019): lines 58, 54.
  3. (Correll et al., 2019): line 50.

Reviewer 3, point 3:

It is necessary to explain what is the originality and novelty of this study in comparison with other studies in the literature.

We thank the reviewer for this comment. This study offers a profile of miRNAs that allows the classification of patients with TRS and SZ with more than 70% accuracy; proposes a new molecular mechanism that could be involved in the development of drug resistance and relates this mechanism to the stress response. All these are novel contributions of this study that clearly differentiate it from the previous ones. We also consider that these points are sufficiently developed throughout the article (Introduction, lines: 72-77; Discussion, lines: 422-425; Conclusion, lines: 533-546).

Reviewer 3, point 4:

The presentation and the objective of the study are too long in introduction (l.61-75). Moreover, results should not appear in the introduction, it looks like an abstract.

This is a very interesting suggestion. However, we believe that the introduction is adequate to understand the concepts and decisions that are developed later in the study. The brief explanation of the methodology is justified by the content of the introduction and we consider that its reduction or elimination would be counterproductive for people unfamiliar with miRNAs or with the RNA-Seq methodology. On the other hand, we do not understand why results should not appear in the abstract. We think the introduction should bring all the necessary concepts to navigate through the rest of the article, so bringing a glimpse of the results helps the reader to integrate the context, the methodology and the results by explaining them in a more extensive way than what is done in the abstract, but in a much reduced way than what is done later in the article. Furthermore, this strategy is rather common, some examples of this can be seen in the following articles:

  1. Langley et al., 2002 (10.1093/emboj/21.10.2383).
  2. Zovoilis et al., 2011 (10.1038/emboj.2011.327).
  3. Hoss et al., 2016 (10.3389/fnagi.2016.00036).
  4. Alberry et al., 2020 (10.1186/s11689-020-09316-3).

Reviewer 3, point 5:

The authors must put the reference numbers instead of the authors' names in table 2.

We really appreciate this remark, we agree with the reviewer and changed the references in table 2 to numbers.

Reviewer 3, point 6:

The figures (1, 2, 3) should be modified to be more visible and axes must be specified.

We appreciate this remark, we moved Figure 1 to the supplementary material at the editor's request, we added the axes labels to Figure 2 and increased the size of figure 2 and 3.

Reviewer 3, point 7:

Some discussion parts need to be modified. Be careful to distinguish between the results and discussion parts. In sub-section 4.3, a discussion related to previous studies is missing. Titles 4.3 and 4.4 should be changed. The molecular model (4.5) could be integrated into the results.

We believe this comment conflicts with the point 2 of reviewer 1. We agree with reviewer 1 that the results are well defined, and we think that they are perfectly separated from the discussion parts. We would appreciate it if reviewer 3 could give us a little more detail on this comment.

Following the reviewer suggestion, we added two references to section 4.3 (Haddad et al., 2014; Valenstein et al., 2004).

We changed the titles 4.3 and 4.4.

We are very thankful for this encouraging comment, but we think this model should be in the discussion as it was not a direct result of our data but a post-hoc interpretation.

Reviewer 3, point 8:

The authors have to check the references, some have to be redone or completed (i.e. 8, 20).

We thank the reviewer for his careful observations. We reviewed in detail the references put using the Zotero add-on for Word with the MDPI citation style. Reference 8 was changed accordingly. On the other hand, citation 20 corresponds to a code repository (GitHub)(https://github.com/sing-group/my-brain-seq) unfortunately, MDPI style does not allow to add any url or extra reference to this kind of citation. However, we agree with the reviewer that the reference is incomplete. We will consult this with the associate editor.

Again, thank you very much for improving the exposition of our work with your thoughts.

Sincerely yours,

Roberto Agís, Hugo López, and Daniel Pérez

Round 2

Reviewer 1 Report

NA

Author Response

Reviewer 1

Round 2:

“English language and style are fine/minor spell check required”

The text was checked for errors.

Sincerely yours,

Roberto Agís, Hugo Fernández and Daniel Pérez

Reviewer 3 Report

Figure 1 should be modified to improve the quality of the figure. Subfigures A, B and D are not visible enough.

Author Response

Reviewer 3

Reviewer 3, point 1:

“Figure 1 should be modified to improve the quality of the figure. Subfigures A, B and D are not visible enough.”

We appreciate this comment, we made subfigures A, B, C and D of Figure 1 more visible: in subplot A, we significantly increased the size of the heatmap and rearranged its legend position to fit with the new heatmap size; in subplot B, we individually increased the size of each subplot and improved the title legibility; in subfigure C, we increased the dendrogram size to make branches and labels more visible; finally in subfigure D, we significantly increased the dendrogram size. We also repositioned the chart letters (A, B, C, D) to correctly fit with the new sizes, increased the figure height and improved the resolution of the png image to facilitate the visualization of figure 1 in digital format.

In addition, the text was checked for errors.

Sincerely yours,

Roberto Agís, Hugo Fernández and Daniel Pérez
